# Does Physical Exercise Enhance the Immune Response after Vaccination? A Systematic Review for Clinical Indications of COVID-19 Vaccine

**DOI:** 10.3390/ijerph20065183

**Published:** 2023-03-15

**Authors:** Luca Barni, Elio Carrasco-Vega, Matteo Olivieri, Alejandro Galán-Mercant, Serena Guiducci, Felice Picariello, Manuel González-Sánchez

**Affiliations:** 1Department of Physiotherapy, Faculty of Health Sciences, University of Málaga, 29071 Málaga, Spain; lucabarnistudio@gmail.com (L.B.); eliotafad@hotmail.com (E.C.-V.); mgsa23@uma.es (M.G.-S.); 2IMT School for Advanced Studies, 55100 Lucca, Italy; 3Institute of Biomedicine of Cádiz (INIBICA), 11009 Cádiz, Spain; 4MOVE-IT Research Group, Department of Nursing and Physiotherapy, Faculty of Health Sciences, University of Cádiz, 11009 Cádiz, Spain; 5Department of Experimental and Clinical Medicine, Divisions of Internal Medicine and Rheumatology AOUC, University of Florence, 50134 Florence, Italy; serena.guiducci@unifi.it; 6Department of Public Health, University Federico II of Naples, 80131 Naples, Italy; felicepicariello@libero.it; 7Institute of Biomedicine of Málaga (IBIMA), 29010 Málaga, Spain

**Keywords:** vaccination, immune response, physical exercise

## Abstract

Background: Stimulating protective immunity with vaccines appears to be the most promising option for providing widespread moderate to high protection against COVID-19 in people over the age of 18. Regular exercise improves the immune response, transmitting possible benefits against virus infections. The aim of this review is to study the effects of physical activity on vaccine injections, helping to develop new recommendations for COVID-19 vaccination campaigns. Methods: A comprehensive review of the existing literature was undertaken using the Preferred Reporting Items for Systematic Reviews and Meta-Analyses (PRISMA) guidelines. The internal quality of the studies was assessed according to the Physiotherapy Evidence Database (PEDro) scale. The outcomes analyzed were antibody titer, the level of lymphocytes CD4, CD8, InterLeukin 6 (IL6), leukocytes level, the visual analogue scale (VAS) for overall pain rating, arm and forearm circumferences and volume of oxygen (VO2) peak. Results: Fourteen articles were selected for the analysis. The majority of studies were randomized controlled trials (RCT) (*n* = 8) and controlled trials (CT) (*n* = 6). According to PEDro, the ‘fair’ category (*n* = 7) was the most represented, followed by ‘good’ (*n* = 6) and ‘excellent’ (*n* = 1). Physical training showed a positive effect on antibody titers of the vaccine; yet, different variables seem to influence antibody titers: higher new vs. old antigen in the vaccine, higher in younger vs. older individuals, and higher in females vs. males. After exercise, when analyzing variables of direct response to the vaccine, such as the amount of CD4, IL-6 and leukocytes, higher levels were observed in the patients who performed physical exercise compared to the control group. In the same way, better results were observed in physiological variables such as VO2 and limb circumferences, or subjective variables such as pain, which showed better results than the control group. Conclusions: The immune response (antibody titers) depends on age, gender and the intensity of physical activity: long-term protocols at moderate intensity are the most recommended. All of these aspects also have to be carefully considered for the COVID-19 vaccination.

## 1. Introduction

Today, the emergency of SARS-CoV-2 severe acute respiratory syndrome and its complications represent a global public health challenge [1], considering the COVID-19 pandemic has resulted in millions of cases and hundreds of thousands of deaths [2]. The main concern related to individuals who tested positive for SARS-CoV-2 is related to the so-called “Long COVID” or more appropriately “Post COVID-19 Syndrome”, which describes the persistence of specific impairments occurring after COVID-19 disease. For instance, a third of the patients affected by COVID-19 experience persistent fatigue and more than a fifth of individuals show cognitive impairment, which persists for 12 or more weeks following their COVID-19 diagnosis [2]. Other symptoms could be breathlessness, myalgia, weakness, headache, cognitive blunting, etc. [3]. Finally, in patients with health issues before a SARS-CoV-2 infection, an aggravation of pre-existing symptoms has also been noticed [4].

The COVID-19 pandemic and the long-term effects of the disease on the individuals who tested positive represent a burden for the health systems. Several individual and social protective measures, including hand washing, the use of face masks, physical distancing and confinement, are associated with reductions in the incidence of COVID-19 [5]. Confinement was a strategy carried out in many countries that helped contain the spread of the virus. However, it also harmed levels of physical activity and caused an increase in sedentary behaviors [4,6].

Nevertheless, stimulating protective immunity with vaccines appears to be the most promising option for managing future infections as mRNA vaccines and adenovirus vector vaccines may provide widespread moderate to high protection against COVID-19 in people over the age of 18 [7].

A patient who suffers from an infection with the SARS-CoV-2 virus may suffer different symptoms and consequences that affect the patient in different ways, some of which have been described the most: headache, malaise, fever, muscle pain and weakness, loss of smell and taste, and respiratory tract problems, among others. In addition, the different mutations that the virus frequently suffers make it necessary to generate a preventive strategy based on the vaccination of the population in order to protect the population and control the number of possible infections from COVID-19 [8].

Nowadays, with the constant mutating of SARS-CoV-2, the implementation of vaccination is critically important. If, on the one hand, governments and relevant agencies are recommended to accelerate the vaccine campaign, on the other, new SARS-CoV-2 variants (Omicron) more often have immune escape ability [8]. The attention to be paid to the vaccination campaign against COVID-19 is still very high, as Omicron not only has many more variants of the S protein, but mainly affects the upper respiratory tract, while the current vaccines instead protect against symptoms related to the lower respiratory tract (lungs).

The literature has described a direct relationship between immune protection and antibody titers, a variable frequently used to estimate the degree of protection that the patient presents against an infection [9,10].

Researchers showed how being physically active affects the immune system [11]. Current evidence from epidemiological studies shows that leading a physically active lifestyle with regular physical activity positively affects systemic inflammatory activity, reducing the incidence of communicable (e.g., bacterial and viral infections) and non-communicable (e.g., cancer) diseases, limiting or delaying immunological aging [12]. There are several possibilities with some degree of evidence to improve the innate immune response and thus transmit possible benefits against viruses, such as healthy lifestyles including regular exercise and a high level of cardiorespiratory fitness. Considering that physical exercise contributes to generating a potentiation in the response of the immune system and considering that vaccination pursues a specific response of the immune system from the introduction of a weakened pathogen, it would be of interest to analyze what the effect of exercise could be in patients who have been vaccinated in order to identify those strategies that could be extrapolated to patients who receive the COVID-19 vaccine as a strategy to enhance their immune response. No systematic review has been identified that analyzes the effect of physical exercise in patients who have received a vaccine to analyze the immune response compared to subjects who do not perform physical exercise.

The aim of this review is to highlight the state of the art of the effects of physical activity on vaccine injections in terms of antibody titer responses, hematic humoral cell levels and pain. This work could help to develop new recommendations for COVID-19 vaccination campaigns.

## 2. Material and Method

### 2.1. Study Design

This systematic review was prepared and structured following the Preferred Reporting Items for Systematic Reviews and Meta-Analyses (PRISMA) guidelines.

### 2.2. Search Strategy

An extensive review of the literature was performed by two investigators, independent and blind to each other. Articles which focused on physical activity’s effects on vaccination were retained. The studies were collected from the following databases: PubMed, SCOPUS, Cochrane, SciELO, PEDro and CINAHL. Various combinations of Medical Topics Heading (MeSH) terms were used including pandemic, physical activity, therapeutic exercise, vaccine, immunology, immunoprotection and immunostimulation. The search strategy did not impose any restrictions on the year of publication and articles published up to August 2022 were included.

### 2.3. Inclusion–Exclusion Criteria

Studies which focused on the effects of physical activity after vaccination on immunization were retained. Articles reporting original research were included if the studies were randomized controlled trials (RCT) or controlled trials (CT), participants were older than 18, the intervention protocol provided the injection of a vaccine and the effect of physical activity was studied. Studies had to have objective measures that evaluated the patient’s immune response, for example, antibody titers. Articles had to be published in English, Spanish, French, Portuguese or Italian. The following exclusion criteria were assigned: a score of <4 on the PEDro methodological quality scale and animal tests.

### 2.4. Document Selection Process

After searching the previously indicated databases, an analysis was conducted to identify duplicate documents. Using the inclusion and exclusion criteria, the results were first filtered by reading the titles and abstracts. In this filtering, if any of the indicated exclusion criteria were identified in the title or abstract, the document was excluded; otherwise, it was included while waiting to be analyzed with the complete reading of the document. Next, with the selected documents, a reading of the entire document was carried out. The last step was to evaluate the methodological quality using the PEDro scale, and those with a score equal to or greater than four were definitively selected.

The document search and selection process was carried out by two evaluators blinded to each other with 15 years of experience in document selection. The documents had discrepancies, so a third reviewer was consulted for the final decision.

### 2.5. Outcomes

Two independent reviewers examined the retained studies to select the outcomes used. The results were structured according to whether they were related to the patient’s immune response after the vaccine (primary variables of the systematic review) or complementary physiological or subjective variables (secondary variables).

#### 2.5.1. Primary Outcomes

In this sense, the immunological response variables analyzed in this systematic review were CD4/CD8, InterLeukin 6 (IL6) and leukocyte level.

An antibody titer is a blood test that determines the presence and level (titer) of antibodies in the blood in order to investigate the immune reaction triggered by antigens. After vaccination, the titer of the specific antibody should be as high as possible.

The blood levels of CD4/CD8, IL6 and leukocytes arre indices that have been correlated to the antibody response following vaccine administration.

CD4 (T-helper cells, T-suppressor cells, and cytotoxic T-cells) and CD8 (cytotoxic T-lymphocytes) are two types of white blood cells which help the body fight infections. Typically, CD4 count is more critical than CD8 count, but they should increase after vaccination. IL-6 is produced wherever there is inflammation, either acute or chronic. Hence, it is usually used as an inflammatory index. Finally, leukocyte levels help to detect infections AAA.

#### 2.5.2. Secondary Outcomes

Secondary outcomes were the visual analogical scale (VAS) for an overall pain rating, arm and forearm circumferences and VO2 peak.

Pain VAS was analyzed to measure muscle soreness after the vaccine injection and in correlation with the level of physical activity. At the same time, limb circumferences were taken to monitor the amount of swelling in the arm submitted to physical activity or the effects on muscles after long-term protocols. Finally, the VO2 peak allowed the researchers to analyze the maximal aerobic capacity to deliver an appropriate intensity of the physical activity protocol or verify the exercise protocol’s efficacy.

### 2.6. Study Quality Assessment

The methodological quality (internal validity) of the literature was assessed with the Physiotherapy Evidence Database (PEDro) scale [7]. The PEDro scale is used to evaluate the quality of the results obtained in a clinical trial. It has a score which ranges from 1 to 10 and the final score is determined by the presence/absence of specific features in the design of the evaluated study. The higher the score, the higher the validity of the evidence. Two independent reviewers evaluated the quality of the studies previously selected, according to the PEDro scale. If PEDro’s final score did not match, a third reviewer was asked for a final evaluation.

## 3. Results

The selection process (identification, screening, eligibility and inclusion) is represented in Figure 1. After performing the search in the different databases, 486 papers were identified, of which, after eliminating duplicate articles, a total of 331 articles were included to apply the inclusion/exclusion criteria. Among them, 50 articles match the inclusion criteria. The PEDro scale score was not sufficient (<4 points) in 28 articles and 8 studies did not report as main outcomes the ones considered in this review. Finally, 14 articles were selected for the analysis after a detailed reading and the evaluation of internal quality.

The internal validity of the selected studies is reported in Table 1. The majority of studies were randomized controlled trials (RCT) (*n* = 8), while the rest of the studies were trials with a controlled group, but with no randomization (CT) (*n* = 6). PEDro scores revealed a tendency towards the ‘fair’ category (PEDro of score 4–5) (*n* = 7), followed by the ‘good’ category (PEDro score from 6 to 8) (*n* = 6) and only one study wasclassified as ‘excellent’ (PEDro score of 9–10) (*n* = 1). The retained studies’ general characteristics are reported in Table 2. The sample size of the included articles ranged from 158 (Edwards et al., 2010) to 7 (Stelzer et al., 2014). The age of the participants ranged from 19.2 (Wang et al., 2011) to 79.5 (Yang et al., 2007) years with a median age of (median[1stIQR-3rdIQR]) 40 [30–45.5] years. Several interventions (from tai-chi to simple eccentric contraction of the biceps), at different levels of intensity, have been studied by the authors. In addition, the amount of training delivered in each protocol was different, with interventions that lasted from one single session to 10 months (Table 2).

### 3.1. Primary Outcomes

#### 3.1.1. Antibody Titers

The influence of physical training on the hematic level of antibody titers of the vaccine compared to a control group (no physical activity) generally showed a positive effect of physical activity according to different variables: higher in younger vs. older individuals and higher in female vs. male individuals (Table 3).

An improvement in antibody titers was observed 28 days after the reduced dose of vaccine administration: fifty eccentric repetitions of deltoid and bicep contraction seem to improve the antibody titers of the least immunogenic strain of a vaccine (B/Florida) and only in men in the A/Uruguay [13]. The level of exercise (60% vs. 85% vs. 100%) does not affect the amount of the response [13].

After a similar exercise protocol, significantly higher titer responses were found in the female exercise group (6 vs. 20 weeks from the vaccination) compared to the female control group (no exercise); in men (exercise group), a significant reduction in antibody titers was measured compared to control (rest group) [14].

An aerobic exercise protocol three times per week for 12 weeks (3 months) in elderly individuals slowed down the reduction of antibodies circulating in blood compared to controls (no exercise), in two of three antigens contained in the vaccine; young participants have a higher antibody titer compared to older adults [17].

A significant increase in influenza-specific antibody levels (H1N1, H3N2, B/Brisbane, B/Phuket) and neutralization titers was observed both among athletes and controls 20 weeks after vaccination; the increase in neutralizing antibodies towards the influenza A strains was significantly stronger in athletes [19].

After 40 min of moderate-intensity aerobic exercise compared to a control group (no exercise), there was no difference in the increment of antibody titers between the two groups in older men; an enhancement in the immune response (only for the H1N1 strain) in older women who submitted to aerobic exercise was reported compared to the female control group [21].

Yang et al. evaluated the efficacy of specific physical protocols which involved Taiji (20 weeks) compared with control (no exercise). Both groups responded to the vaccine with significant elevations in titer level; the Taiji group revealed higher levels of antibody titers (geometric mean of anti-influenza antibody titers) when compared to the controls; in particular, the Taiji group had significantly higher (compared to pre- vaccination) titers at 20 weeks post-vaccine [24].

#### 3.1.2. CD4/CD8

The immune function after exercise, evaluated with CD4 count, resulted in improvement compared to a control group (no exercise) in five studies out of six (Table 3).

In particular, CD4 count improved in both groups (exercise and no exercise), but the increase was significantly more pronounced in athletes [19]. Another study reported an improvement in CD4 count in the female exercise group compared to the female control group, but no statistical changes were observed in CD8 count [23]. An improvement in CD4 and CD8 count was found in healthy individuals after the adenovirus-specific T-cell expansions followed by one aerobic session [18]. A pre–post significant difference of the CD4 levels was not found between a group practicing a long-term exercise protocol and a control group among patients with HIV [16]; conversely, the CD4 and CD8 level was found to be significantly improved after 8 and 12 weeks of aerobic and non-linear resistance training (3 times/week) in patients with HIV compared to a control [15,26].

#### 3.1.3. IL6

Levels of IL-6 also increase after the administration of a vaccine compared to the sham group; moreover, there was a significant correlation between IL-6 levels 24 h after vaccination and antibody titers (H1N1 strain) after 4 weeks in the exercise group, which was not significant in the control group [21] (Table 3). Among individuals affected by HIV, physical activity correlated with a reduction in the inflammatory markers, such as IL-6 [26]. Finally, after an ultra-endurance race an improvement in the level of IL-6 and other inflammatory markers was observed [22] (Table 3).

#### 3.1.4. Leukocytes

Regarding the levels of leukocytes, only two studies measured this index, finding an improvement in the levels of hematic leukocytes in the adenovirus-specific T-cell expansion group [18] and after an ultra-endurance race [22].

### 3.2. Secondary Outcomes

#### 3.2.1. VAS (0–100/GRADE I–III)

Comparing self-reported arm pain scores, a significant group effect between exercise and control (no exercise) has been highlighted: in the pain injection site, the exercise group showed higher pain scores compared to controls; in general, exercise groups reported higher levels of perceived pain compared to the no exercise group and pain sensation seems higher when the participant moved the arm. No sex differences were found in exercise-induced pain (Table 4).

#### 3.2.2. Upper Arm and Forearm Circumferences

Comparing limb circumference percentage changes from baseline to post-task revealed significant group effects in favor of the exercise groups compared to controls; a significant sex difference for upper arm circumference was found in favor of males compared to females (Table 4).

#### 3.2.3. VO2 Peak

To evaluate the effects of the combination of exercise training and statins in people living with HIV, the exercise + placebo and exercise + statins groups significantly increased the VO2 peak compared with no exercise (+placebo and +statin) groups. In the exercise groups, the improvement in the VO2 peak was statistically significant compared to controls (no exercise) [15,25], even if in one case, no difference between groups was found [25].

## 4. Discussion

This review aimed to provide an overview of the current evidence on the effects of physical activity with respect to the hematic response after vaccine. The included studies were 14 out of 50; among the unselected studies, the internal quality was low in the 78%, with a PEDro score lower than four, suggesting greater attention is required as to how studies are designed to answer related research questions. Furthermore, even if most of the included studies were RCT (57%), the PEDro score ranged between 4 and 5 points mainly (50%), highlighting again the rather low quality of studies on the topic.

The characteristics of the included participants were mixed, as the immune response may vary according to gender and age, and authors selected young/adults and female/male individuals. Finally, the physical activity protocol used to enhance the immune response was varied too, studying the topic both after one single session and after a specific long-term protocol, or in athletes. Furthermore, the analysis showed a lack of agreement on the specific indexes to evaluate and on the measurement unit to use for the immune response after vaccination; in addition, the protocols used to measure the pre–post- results varied. These aspects make comparisons among different studies difficult.

Physical activity, in general, can enhance the immune response to vaccination (antibody titers), but also the immune system, such as the CD4 level [7]. Concerning leukocytes levels, their role after exercise is still unclear [23], as well as in this review, where the data collected were not sufficiently homogeneous to infer a clear indication. On the other hand, IL-6 typically increases with physical activity [27], as reported also in this analysis.

The immune response after a vaccine is correlated to the specific strain: the pattern of response seems to be more pronounced with new antigens in the vaccine, so that the immune system reacts more “vigorously” [5]. Nonetheless, the amount of the immune response (higher or lower antibody titers) depends also on other factors such as age, gender and the intensity of physical activity.

### 4.1. Age and Immune Response

The antibody titers after vaccine currently analyzed in this article were generally in agreement with the literature showing how the immune response is actually higher in young people in comparison to older individuals [28]; interestingly, the current analysis showed how physical activity practiced with regularity can reduce the decrement of the antibodies in older adults [17].

### 4.2. Gender and Immune Response

Additionally, gender seems to influence the pattern of response after vaccination, generally showing higher antibody titer responses in women compared to men, as discussed by others [29]. Sex differences in the responsivity to some vaccine strains are well documented: sex hormones (estradiol, progesterone, and testosterone) may play a major role in differences between male and female immune responses [29]. Interestingly, sex differences, probably mediated by immune response, have been highlighted also in COVID-19 infections. In particular, there is a biological sex difference in the expression and regulation of angiotensin-converting enzyme 2, which is the main receptor used by COVID-19 to enter cells [12]. Although females tend to experience less severe disease in response to viral infection, these infections are thought to contribute to higher rates of autoimmune disease observed in this population, as the “X” chromosome encodes several genes involved in innate and adaptive immune function [11]. Finally, sex differences can be found in the response to physical activity, even if conclusions on the topic are yet to be confirmed [14]. Nonetheless, some attempts show how muscle damage could occur similarly in men and women, but the inflammatory response (i.e., smaller increase in leukocytes) was reduced in women [30].

### 4.3. Physical Activity Intensity and Immune Response

Physical protocols used by the authors were very different from each other, and even if the effect of exercise level seems not to influence the immune response [13], differences between long-term protocols have been raised by the analysis. In particular, regular exercise (i.e., moderate-intensity exercise, Taiji, etc.) practice for at least 12 weeks seems to stimulate the immune system and also the response after vaccination [7]; in addition, in athletes, the response of the immune system appears to be higher [19,31].

### 4.4. Secondary Outcomes

The results related to secondary outcomes suggest that the subjective perception of pain improves with physical activity (exercise group) compared to the control group (no exercise). The pain perceived correlates with the intensity of the movement itself; therefore, the greatest increases in pain in the arm were observed in participants exposed to the highest intensity exercise. Concerning limb circumferences, men showed greater exercise-induced increases in upper arm circumference than women [14]. The gender difference could be due to sex hormones and their homeostasis, even if the literature is not clear about the mechanism underlying these phenomena [30]. For longer physical activity protocols, authors also reported a reduction in the subcutaneous body fat and in neck, abdomen, and waist circumferences, demonstrating the beneficial effects of regular exercise.

A systematic protocol on how to use VO2 peak index between different studies was not found: it was used both to verify the maximum exercise threshold before the protocol and to compare metabolic responses in the exercise groups compared to the control group (no exercise). It also showed positive results, as VO2 peak increased after an exercise protocol. The former showed a positive correlation between the improvement of the hematic level of CD4 and the improvement in the VO2 peak [25], demonstrating how the index should be used probably to guide the physical activity intensity, but also to monitor the results of the protocol.

## 5. Study Limitations

This systematic review is the first to highlight the correlation between physical activity and vaccine. Nevertheless, limitations can be found in the study protocol. In particular, the articles included in the analysis were consulted as aggregated data, so that it was impossible to have access to the specific database of the single studies. Secondly, even if, to the best of our capacity, articles written in five languages were selected for this analysis, there could be other studies written in other languages which are useful for implementing the considerations on the topic.

## 6. Conclusions

The current analysis showed how the immune response after vaccination and physical activity compared to no exercise leads to a better immunization response, even if several aspects have to be considered. First of all, sex differences in immune responses both to vaccination (higher in women) and to viral infections (less severe in women) must be considered. Then, the different effects of vaccination in older adults, in which the immune response is normally lower than that young individuals. Nonetheless, moderate-intensity physical activity, practiced with consistency, resulted in the best outcomes as it enhanced the immune system and it could also raise the vaccination response, both in men and women and in older and younger individuals. It could be possible to induce a higher immune response after vaccination with specific physical activity protocols practiced on the day of the vaccination dose, even if this could lead to higher levels of pain perceived.

All of these aspects also have to be carefully considered for COVID-19 vaccination, promoting physical activity for the general population and maybe specific protocols to use to enhance the immune response on the day of the COVID-19 vaccination.

## Figures and Tables

**Figure 1 ijerph-20-05183-f001:**
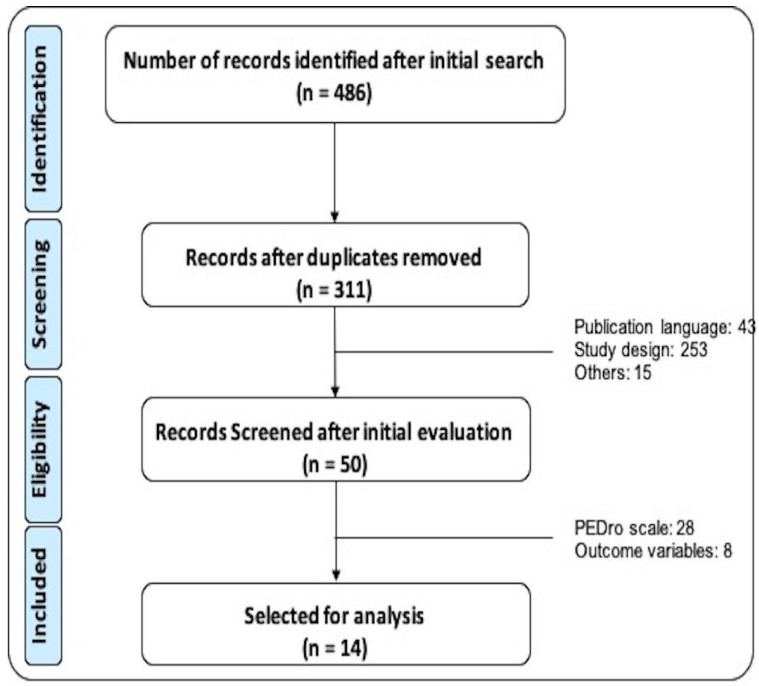
Search, filtering and selection flowchart of the documents included in this systematic review.

**Table 1 ijerph-20-05183-t001:** PEDro score.

Author (Year)	Type of Study	Item 2	Item 3	Item 4	Item 5	Item 6	Item 7	Item 8	Item 9	Item 10	Item 11	Total
Edwards (2010) [13]	RCT	1	0	0	0	0	0	1	1	1	1	5
Edwards (2007) [14]	RCT	1	0	0	0	0	0	1	0	1	1	4
Ezema (2015) [15]	RCT	1	0	1	0	0	0	1	1	1	1	6
Gomes (2010) [16]	CT	0	0	0	0	0	0	1	1	1	1	4
Kohut (2004) [17]	CT	1	0	0	0	0	0	1	0	1	1	4
Kunz (2018) [18]	RCT	0	1	1	1	1	0	1	1	1	1	8
Ledo (2019) [19]	CT	0	0	1	0	0	0	1	1	1	1	5
Miles (2002) [20]	CT	1	0	0	1	0	0	1	1	1	1	6
Ranadive (2014) [21]	RCT	1	1	1	0	0	0	1	0	1	1	6
Stelzer (2014) [22]	CT	1	0	0	1	0	0	1	0	1	1	5
Wang (2011) [23]	RCT	1	0	1	0	0	0	1	1	1	1	6
Yang (2007) [24]	CT	0	0	0	0	0	0	1	1	1	1	4
Zanetti (2019) [25]	RCT	1	1	1	1	1	0	1	1	1	1	9
Zanetti (2016) [26]	RCT	1	0	1	0	0	0	1	1	1	1	6

**Table 2 ijerph-20-05183-t002:** Structural characteristics of the studies included in this systematic review.

Author	Sample Size		Age (Years)	Intervention	Session Frequency	Outcome Variables
Edwards et al. (2010) [13]	160	G1 (exercise 60%): 39 G2 (exercise 85%): 40 G3 (exercise 110%): 40 G4 (control): 39	G1: 20.3 ± 1.2 G2: 20.0 ± 1.2 G3: 20.8 ± 2.0 G4: 20.8 ± 2.0	Experimental groups (G1, G2; G3): concentric movements for the non-dominant arm at 60%/85%/110%, respectively, of the concentric repetition maxima. Control group (G4): resting for 25 min.	only one session performed	Limb circumference; pain (VAS); sensation of exertion (BORG); blood analysis (anti-influenza antibodies, IL-6, CK)
Edwards et al. (2007) [14]	60	G1 (exercise): 40 G2 (control): 20	Man: 20.1 ± 1.64 Woman: 20.6 ± 2.55	G1: eccentric contraction of the biceps brachii and deltoid muscles of the non-dominant arm at 85% of single repetition concentric maxima. G2: remained resting quietly for a further 25 min.	only one session performed	Pain sensation (McG- ill Pain Questionnaire); upper arm circumference; blood sampling (anti-influenza antibodies, cell-mediated antigen-specific immunity, IFN-*γ*); other scales (LESS, USQ, PSS, GHQ-28, Whitehall II study).
Ezema CI (2015) [15]	33	G1 (exercise): 17 G2 (control): 16	G1: 40.1 ± 9.7 G2: 32.5 ± 10.4	G1: EG received exercised of moderate intensity of between 60% and 79% of their heart rate (HR) reserve as recommended by ACSM quantified by jogging on a treadmill. G2: conventional therapy only (antiretroviral therapy)	G1: The exercise session was increased from 45 min in the first 2 weeks of training and leveled up to 60 min throughout the remaining part of the training. Exercise sessions of 3 times/week for 8 weeks	Systolic blood pressure (SBP), diastolic blood pressure (DBP), maximum oxygen uptake (VO2 max) and CD4 cell count
Gomes et al. (2010) [16]	29	HIV+ G1 (exercise): 19 G2 (control): 10	G1: 46.0 ± 3.0 G2: 43.0 ± 5.0	G1: aerobic (30′), strengthening (50′), flexibility activity (10′). G2: normal physical activity.	3 sessions per week. For 12 weeks.	Life Satisfaction Index (LSI); blood analysis (CD4 count total and relative).
Kohut et al. (2004) [17]	27	G1 (exercise): 14 G2 (control): 13	G1: 73.1 ± 5.6 G2: 70.3 ± 5.6	G1: supervised aerobic exercise class. G2: continue their current exercise program.	25–30 min. 3 sessions a week. For 10 months.	Blood samples (H1N1, H3N2, B titers, Granzyme B assay).
Kunz HE et al. (2018) [18]	24	G1 (acute exercise group): 14 G2 (AdV-specific T-cell expansions): 10	G1: 31.3 ± 4.6 (26–43) G2: 32.8 ± 5.3 (26–43)	G1–G2 first intervention: performed a submaximal, discontinuous, incremental cycling protocol on a stationary indoor cycle ergometer (Velotron, Racermate, Seattle, WA, USA) to determine the individual blood. G1–G2 second intervention: The second visit consisted of a 30 min steady-state cycling protocol at a power output 10–15% above the power output at the individual blood lactate threshold	G1 and G2 sessions separated by at least two days but not more than 2 weeks G2: sessions separated by at least two days but not more than 2 weeks	Immune cellular responses to acute exercise. Total and differential leukocyte counts before (PRE) and after (POST): NK cells, T-cells, CD4+ and CD8+ T- cells, and CD4+ and CD8+ T-cell. The effects of exercise on AdV-specific T-cells
Ledo a et al. (2019) [19]	71	G1 (athletes): 46 G2 (controls): 25	G1: 23.2 ± 7.7 G2: 22.8 ± 4.1	G1: training for international or at least national level G2: leisure sport	G1: training 5 days a week. One session refers to a time frame between 1.5–4 h G2: leisure sport with no more than 2 training sessions per week	CD-4T-cells; leukocytes; lymphocytes; cytokine profile; IL-2; IFNy; TNFalfa; Interferone pain at the injection (duration and intensity).
Miles et al. (2002) [20]	10	G1 (exercise): 6 G2 (control): 4	G1: 32.8 ± 6.9 G2: 28.8 ± 7.5	G1: perform a 60 min run G2: time-matched control	only one session was performed	Aerobic capacity test (VO2 peak); lymphocyte quantification and phenotyping; NK cell cytotoxic activity; isolation of mononuclear cells and RNA; Perforin mRNA.
Ranadive et al. (2014) [21]	55	G1 (exercise): 28 G2 (control): 27	G1: 66.0 ± 0.9 G2: 67.0 ± 0.8	G1: Visit 1 (V1) before treatment (VO2 measurement); at least 7 days after V1, 40 min treadmill aerobic exercise and then anti-influenza vaccination. G2: V1 before treatment; at least 7 days after V1, sham injection.	(V1): first visit (VO2 peak, anthropometrics); (V2): treatment/no treatment + injection; (V3): blood samples after 24 h (inflammatory markers); (V4): blood samples after 48 h (inflammatory markers); (V5): blood samples after 4 weeks (efficacy markers).	Aerobic capacity test (VO2 peak); anthropometrics; inflammatory markers (CRP; IL-6); efficacy marker (specific anti-influenza antibody responses).
Stelzer I et al. (2014) [22]	7	G1: 7	G1: 39.6 ± 7.8	Ultra-endurance cycling racers took part in this study. The exercise workload for each participant was 550 km and a 7000 m altitude difference within 4 days, with 8 h of competition alternating with 8 h of rest.	G1: Single competition	Lactate; leukocytes; neutrophil granulocytes abs; monocytes abs; lymphocytes abs; eosinophil granulocytes abs; basophil granulocytes abs; erythrocytes; hemoglobin; hematocrit; platelets; CD34+/CD45; BFU-E; CFU-GM; MMP-9, TIMP-1; cortisol levels; IL-6; fibrinogen; CK; CK-MB; HSCTNT; NT-pro-BNP; myoglobin; LDH; AST; ALT.
Wang et al. (2011) [23]	60	G1 (TCC): 30 G2 (control): 30	G1: 19.2 ± 1.3 G2: 19.5 ± 2.1	G1: incorporating elements of balance, postural alignment and concentration, under the guidance of a master: 10 min warm-up; 30 min of practice; 5 min cool-down. G2: normal physical activity.	45 min. 5 times per week. For 12 weeks.	URTI ratio; blood analysis (IgG, IgA, IgM, IFN-*γ*; IL-4, IL-12) + (CD3, CD4+, CD8+).
Yang et al. (2007) [24]	50	G1 (TQ): 27 G2 (control): 23	G1: 79.5 ± 1.9 G2: 74.5 ± 1.6	G1: Qigong and Taiji form practice G2: continue routine activities for 20 weeks	60 min. 3 sessions a week. For 20 weeks.	Sleep quality (PSQI); blood samples (anti-influenza antibody titer; H1N1; H3N2;
Zanetti H.R (2019) [25]	82	G1 (placebo): 21 G2 (statin): 21 G3 (placebo + exercise):20 G4 (statin + exercise) 2 0 B)	G1: 44.8 ± 10.7 G2: 43.0 ± 9.8 G3: 39.9 ± 10.1 G4: 40 ± 10.8	G1: placebo G2: 10 mg calcium rosuvastatin G3: placebo + 12 week exercise training, intervention composed of periodized nonlinear resistance training (RT) and periodized polarized training on treadmill (PT) G4:10 mg calcium rosuvastatin +12 week exercise training, 3 times per week intervention composed of periodized nonlinear resistance training (RT) and periodized polarized training on treadmill (PT)	G1: one pill per day G2: one pill per day G3: one pill per day + 12 week exercise training, 3 time per week G4:10 one pill per day + 12 week exercise training, 3 time per week	Body composition, lipid and inflammatory profile (interleukin), cardiovascular disease marker, doppler ultrasound, muscle strength (KG), cardiorespiratory fitness (VO2 max).
Zanetti H.R. (2016) [26]	30	G1 (non-linear resistance training): 15 G2 (control): 15	G1: 41.5 ± 11.4 G2: 40.7 ± 8.8	G1: 12 weeks of intervention with resistance exercise consisting of six exercises that emphasize large muscle groups: squat, bench press, hamstring curl, frontal pull, seated calf raise, and shoulder press. Perform each series to concentric failure. G2: maintain daily habits.	G1: 3 times per week	Body fat, subcutaneous fat (mm), body circumferences (cm), muscular strength (KG) and inflammatory profile (Cytokines (pg/mL), T cells (cells/mm^3^), Viral load (copies/mm^3^)

**Table 3 ijerph-20-05183-t003:** Results of the hematic variables analyzed in the different studies included in this systematic review.

Measurement Tool	Article	Group	Unit of Measure	First Measurement	End Measurement
*Antibody titers*	Edwards (2010) [13]	G1 (60%)	A/Brisbane A/Uruguay B/Florida	22 (16–32) 17 (14–21) 197 (128–303)	425 (287–632) 116 (76–177) *(M) 1433 (1127–1821) *
		G2 (85%)	A/Brisbane A/Uruguay B/Florida	19 (14–27) 15 (12–18) 180 (122–264)	323 (221–473) 129 (79–212) *(M) 813 (627–1056) *
		G3 (110%)	A/Brisbane A/Uruguay B/Florida	15 (12–19) 14 (11–17) 270 (200–370)	440 (286–675) 61 (38–99)—*(M) 1220 (951–1565) *
		G4 (control)	A/Brisbane A/Uruguay B/Florida	16 (13–19) 17 (13–23) 388 (300–501)	333 (232–476) 104 (64–169) 1220 (1046–1423)
		Overall	A/Brisbane A/Uruguay B/Florida	20 (17–23) 16 (14–18) 246 (206–294)	385 (320–463) 100 (79–126) 1152 (1028–1290)
	Edwards (2007) [14]	G1 (exercise) men	A/New Caledonia A/Wyoming B/Jiangsu	≃1900 ≃400 ≃180	≃1300 * ≃330 * ≃100 *
		G1 (exercise) women	A/New Caledonia A/Wyoming B/Jiangsu	≃1500 ≃780 ≃410	≃1000 * ≃550 * ≃230 *
		G2 (control) men	A/New Caledonia A/Wyoming B/Jiangsu	≃3000 ≃750 ≃390	≃1500 ≃580 ≃250
		G2 (control) women	A/New Caledonia A/Wyoming B/Jiangsu	≃800 ≃600 ≃300	≃400 ≃330 ≃200
	Kohut (2004) [17]	G1 (exercise)	A H1N1 A H3N2 Type B	7.30 ± 0.51 6.23 ± 0.32 5.93 ± 0.24	6.95 ± 0.54 * 6.45 ± 0.25 5.32 ± 0.27
		G2 (control)		6.08 ± 0.34 5.67 ± 0.30 5.74 ± 0.32	5.70 ± 0.33 6.54 ± 0.30 5.73 ± 0.28
		G3 (young)		7.4 ± 0.74 4.94 ± 0.26 7.68 ± 0.45	6.73 ± 0.94 6.68 ± 0.49 6.73 ± 0.29 *
	Ledo (2016) [19]	G1 (exercise)	A H1N1 A H3N2 B/Brisbane B/Phuket	≃6000 ≃640 ≃160 ≃400	≃1000 * ≃640 ≃100 * ≃200
		G2 (control)		≃2560 ≃640 ≃160 ≃360	≃800 * ≃400 ≃100 ≃160
	Ranadive (2014) [21]	G1 (exercise) men	H1N1 H3N2	≃1.5 ≃4.0	≃2.5 * ≃5.0 *
		G1 (exercise) women		≃1.0 ≃3.0	≃4.0 * ≃5.0 *
		G2 (control) men		≃2.5 ≃3.0	≃4.5 * ≃5.5 *
		G2 (control) women		≃2.2 ≃3.8	≃3.5 * ≃5.0 *
		G1 (exercise) men + women	B-Brisbane	≃1.0	≃2.5 *
		G2 (control) men + women		≃1.0	≃3.0 *
	Yang (2007) [24]	G1 (exercise)	Geometric Mean Anti-Influenza Antibody titers	≃2.6 *	≃2.4 *
		G2 (control)		≃1.5	≃0.8
*CD4*	Ezema (2015) [15]	G1 (exercise)	cells/mm^3^	516.00 ± 256.49	656.27 ± 189.17 *
		G2 (control)		492.27 ± 229.86	510.93 ± 226.14
	Gomes (2010) [16]	G1 (exercise)	cells/mm^3^	503.9 ± 55.0	565.6 ± 72.1
		G2 (control)		462.2 ± 39.7	398.1 ± 69.4
		G1 (exercise)	%	20.3 ± 2.1	23.5 ± 2.0
		G2 (control)		20.8 ± 4.0	21.7 ± 3.5
	Kunz (2018) [18]	G1 (exercise)	cells/μL	810 ± 256	1018 ± 344 *
			%	55.3 ± 11.3	46.4 ± 12.6 *
	Ledo (2019) [19]	G1 (exercise)	fold increase	≃3.8 *	≃1.5 *
		G2 (control)		≃2.5	≃1.0 *
	Wang (2011) [23]	G1 (exercise)	%	≃35%	≃40% *
		G2 (control)		≃34%	≃35%
	Zanetti (2016) [26]	G1 (exercise)	Δ pre-post cells/mm^3^	64.4 ± 38.5 *	
		G2 (control)		6.2 ± 59.5	
*CD8*	Kunz (2018) [18]	G1 (exercise)	cells/μL	491 ± 201	840 ± 370 *
			%	32.2 ± 7.4	36.8 ± 10.1 *
	Wang (2011) [23]	G1 (exercise)	%	≃30%	≃30%
		G2 (control)		≃29%	≃31%
	Zanetti (2016) [26]	G1 (exercise)	Δ pre-post cells/mm^3^	186.6 ± 78.7 *	
		G2 (control)		3.8 ± 36.1	
*IL-6*	Ranadive (2014) [21]	G1 (vaccine)	Δ pre-post (24 h)	≃0.8 *	
		G2 (sham)		≃0.2	
		G3 (vaccine)	Δ pre-post (48 h)	≃0.3 *	
		G4 (sham)		≃0.1	
	Zanetti (2016) [26]	G1 (exercise)	cells/mm^3^ Δ pre-post	−2.3 ± 0.4 *	
		G2 (control)		0.09 ± 0.3	
	Stelzer (2014) [22]	G1 (exercise)	pg/mL	2.8 ± 2.9	5.7 ± 3.9 *
*Leukocytes*	Kunz (2018) [18]	G1 (exercise)	x103/μl	6.1 ± 1.5	10.3 ± 2.2 *
	Stelzer (2014) [22]	G1 (exercise)	G/L	5.12 ± 0.75	7.39 ± 1.29 *

Table legend: “*” *p* < 0.05.

**Table 4 ijerph-20-05183-t004:** Results of the pain and physical performance variables analyzed in the studies included in this systematic review.

Measurement Tool	Article	Group		Baseline	End Protocol
*VAS (0–100 nrs; I–III grade)*	Edward (2010) [13]	G1 (60%)	Resting Pain	0	8.33 ± 15.55
			Movement Pain	0	15.08 ± 15.6
		G2 (85%)	Resting Pain	0	9.84 ± 13.80
			Movement Pain	0	25.25 ± 18.47
		G3 (110%)	Resting Pain	0	13.64 ± 17.37
			Movement Pain	0	25.74 ± 21.52
		G4 (control)	Resting Pain	0	1.37 ± 1.24
			Movement Pain	0	2.66 ± 2.91
	Edward (2007) [14]	G1 (exercise)	Men	0	42—12 (6 h)—42 (24 h)
			Women	0	40—15 (6 h)—30 (24 h)
		G2 (control)	Men	0	2—1 (6 h)—3 (24 h)
			Women	0	8—2 (6 h)—12 (24 h)
	Ledo (2019) [19]	G1 (athletes)	No Pain	100%	≃27%
			Grade I	0	≃13%
			Grade II	0	≃50%
			Grade III	0	≃10%
		G2 (control)	NP	100%	≃48%
			Grade I	0	≃21%
			Grade II	0	≃31%
			Grade III	0	0%
*Upper arm and forearm circumferences*	Edward (2010) [13]	G1 (60%)	Arm	-	0.64 ± 1.49
			Forearm	-	1.37 ± 1.61
		G2 (85%)	Arm	-	0.92 ± 2.56
			Forearm	-	2.52 ± 2.06
		G3 (110%)	Arm	-	1.85 ± 2.49
			Forearm	-	3.17 ± 3.85
		G4 (control)	Arm	-	-0.15 ± 0.45
			Forearm	-	-0.20 ± 1.12
	Edward (2007) [14]	G1 (exercise)	Men	28.2	≃29; ≃28.7 (6 h)
			Women	26.8	≃26.8; ≃ 27 (6 h)
		G2 (control)	Men	28.3	≃27.9; ≃28 (6 h)
			Women	26.9	≃27; ≃26.8 (6 h)
	Zanetti (2016) [26]	G1 (NLRT) Δ pre-post	Neck	−0.7 ± 0.5	
			Chest	0.83 ± 1.3	
			Abdomen	−1.3 ± 1.9	
			Waist	−1.4 ± 1.9	
			Hip	−0.5 ± 2.3.	
		G2 (control) Δ pre-post	Neck	−0.2 ± 0.5	
			Chest	0.5 ± 0.5	
			Abdomen	1 ± 0.6	
			Waist	0.2 ± 0.4	
			Hip	0.5 ± 0.6	
*VO2 peak (mL kg^−1^ min^−1^)*	Ranadive (2014) [21]	G1 (exercise)	-	25.90 ± 1.20	-
		G2 (control)	-	25.14 ± 1.29	-
	Miles (2002) [20]	G1 (exercise)	-	55.0 ± 2.9	-
		G2 (control)	-	49.6 ± 3.5	-
	Zanetti (2019) [25]	-	-	Initial Values	Δ pre-post
		G1 (placebo)		30.9 ± 8.6,	0.03 ± 1.7–0.8 to 0.8
		G2 (statin)		33.3 ± 3.6,	0.2 ± 2–0.3 to 1.3
		G3 (placebo + exercise)		33.5 ± 3.9	10.4 ± 3.2–8.8 to 12.1
		G4 (statin + exercise)		32.4 ± 2.7.	11.1 ± 4.3–8.9 to 13.2
		Overall	-	32.7 ± 3.4	-
	Ezema (2015) [15]	G1 (exercise)	-	23.00 ± 2.54	30.87 ± 4.47
		G2 (control)	-	24.00 ± 2.54	23.87 ± 2.65.

## Data Availability

Data are available under rationated request.

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
