# Peer review of "Does Physical Exercise Enhance the Immune Response after Vaccination? A Systematic Review for Clinical Indications of COVID-19 Vaccine"

_ijerph, 2023, doi:10.3390/ijerph20065183_

Round 1

Reviewer 1 Report

It is an interesting systematic review of the effects of exercise on the immune response to influenza vaccination. Although the topic is very interesting, it has important methodological flaws that must be corrected to be considered for publication.

I suggest authors rewrite the article and meet PRISMA requirements.

Abstract

-There are some abbreviates not defined in the abstract. E.g., VAS, CT, RCT.

-I suggest improving the writing of the results. It is unclear whether the changes in CD4 (and other variables) are related to exercise itself or exercise in the context of vaccination.

Introduction

-The paragraph between lines 50 and 56 is decontextualized from the general idea. According to published other articles, I suggest adding the antecedents derived from confinement [sedentary behavior] as secondary effects of COVID-19.

-The paragraph between lines 62  and 69 should be rechecked. It is not easy to understand.

-The hipótesis is scarcely justified in the introduction. Why is it plausible to suppose that exercise could help to improve immune response in the context of viral vaccination?

- According to the title, the review is focused on anti-influenza vaccination; however, in the objectives, COVID-19 vaccination is mentioned. What it is a crucial incongruence. More specific antecedents of influenza and exercise should be added.

-In the background, it is unclear if the study focus is related to acute or chronic exercise. It should be explained.

- In the background, it should be provided antecedents regarding variables. Why is antibody titer considered a primary outcome? Why is it essential for the focus of the study? Explain.

Method

-Review should be rewritten according to the PRISMA statement.

-To my understanding, Medline is the same as Pubmed.

-The medical subject headings related to the viral disease are about COVID-19. Why not add any term regarding influenza? If the review is about influenza vaccination, related terms must be added.

-For methodological rigor, If authors decided to add terms related to COVID but not related to influenza vaccination, the publication limit to consider primary articles should be the onset of the COVID-19 pandemic [november 2019].

-Why was a low score on the PEDRO scale considered an exclusión criterion?  

-If an inclusion criterion is having studied physical activity, it can not be an exclusion criterion for any article that has not studied physical activity. It should be checked.

-The eligibility criteria of primary articles are unclear. It should consider study design, context, subjects, interventions, and outcomes. That should be presented with more precisión in the method.

-In a systematic review of physical activity or exercise is mandatory to consider the duration of training as eligibility criteria [acute response vs. chronic adaptation to exercise]. It is not mentioned in the method. It should be considered in the method and introduction [explay why, for what, and so on].

-It is unclear how the screening sequence of primary articles will be performed.

-The criteria of abstract exclusion should be added.

- in the method, the description of outcomes looks like this study were an experimental design. It should be corrected.

-It is not clear why Pain was included as a secondary variable. It should explain its relationships with the focus of the study [in the introduction, preferably].

Results

-Figure 1 was not included.

-Is the number 486 titles or abstract? It is not clear.

-The sequence of identification of primary articles is unclear. It should be according to PRISMA.

Reviewer 2 Report

This is a valid, well presented systematic review addresses the issue of the effect of physical exercise to the immune response after the influenza vaccination and its findings indicate according to the authors in the abstract that “The immune response (antibodies titres) depends on age, gender and the intensity of physical activity: long-term protocols at moderate intensity are the most recommended”.

Their selection process ended up with 14 articles, and as the authors say they had difficulty in assessing and processing the different data form the different study protocols. As the authors state in line 338: “this systematic review is the first to highlight the correlation between physical activity and influenza vaccine” and is indeed a valid reference to scientists working in this field, but it fails to provide adequate justification on other the statement the authors make; regarding the results of their research and how they can be considered for the COVID-19 vaccination programs. This claim is made in the title of the article as well, when all the analysis is performed on data from research papers for physical exercise and whether it enhances the immune response after the influenza vaccination, not COVID-19 vaccination. These are different types of vaccines, produced for different viruses and the authors do not provide enough evidence for their claim. They have to better justify or explain the correlation they suggest or remove this statement completely. In particular, the authors should explain why they are extrapolating the results from the influenza vaccination studies to suggest modification on the COVID-19 vaccination existing recommendations.  Even though, their work could be of assistance to people who are currently conducting similar research for COVID-19 vaccines and physical exercise effect on the immune response, in order for example to give them some points to consider when designing their experiments, I fail to see how they can conclude that could be directly used in altering the vaccination recommendations.  

Round 2

Reviewer 2 Report

This revised version, answered previous questions and observations. This systematic review could contribute new information to the field.